# Physiological response of anthocyanin synthesis to different light intensities in blueberry

Xiaoli An[1], Tianyu Tan[2], Zejun Song[1], Xiaolan Guo[3], Xinyu Zhang[1], Yunzheng Zhu[1], Delu Wang🔵[1]*

1 College of Forestry, Guizhou University, Huaxi, Guiyang, Guizhou, China, 2 Forestry Bureau of Kaili, Kaili, Guizhou, China, 3 College of Life Sciences, Huizhou University, Huizhou, Guangdong, China

* deluwang23@aliyun.com

**Data Availability Statement:** All relevant data are within the paper and its Supporting information files.

**Funding:** Delu Wang, grant number 31760205, National Natural Science Foundation of China. The

## Abstract

Fruit color is an important economic character of blueberry, determined by the amount of anthocyanin content. Anthocyanin synthesis within the blueberry fruits is significantly affected by light. To reveal the physiological response mechanism of anthocyanin synthesis in blueberry fruits in different light intensities, four light intensities (100% (CK), 75%, 50% and 25%) were set for the 'O'Neal' southern highbush blueberry as the experimental material in our study. The relationship between endogenous hormones content, associated enzyme activities, and variations with the anthocyanin content in blueberry fruits under various light intensities during the white fruit stage (S1), purple fruit stage (S2), and blue fruit stage (S3) were studied. The results showed that adequate light could significantly promote anthocyanin synthesis in blueberry fruits ($P < 0.05$). Blueberry fruits had an anthocyanin content that was 1.76~24.13 times higher under 100% light intensity than it was under non-full light intensity. Different light intensities significantly affected the content of endogenous hormones and the activity of associated enzymes in anthocyanin synthesis pathway ($P < 0.05$). Among them, the JA (jasmonic acid) content and PAL (phenylalanine ammonia lyase) activity of fruits under 100% light intensity were 2.49%~41.83% and 2.47%~48.48% higher than those under other light intensity, respectively. And a significant correlation was found between the variations in anthocyanin content in fruits and the content or activities of JA, ABA (abscisic acid), ETH (ethylene), $GA_3$ (gibberellin 3), IAA (indoleacetic acid), PAL, CHI (chalcone isomerase), DFR (dihydroflavonol reductase) and UFGT (UDP-glucose: flavonoid 3-glucosyltransferase) ($P < 0.05$). It indicated that 100% light intensity significantly promoted anthocyanin synthesis in blueberry fruits by affecting endogenous hormones content and associated enzyme activities in the anthocyanin synthesis pathway. This study will lay a foundation for further research on the molecular mechanism of light intensity regulating anthocyanin synthesis in blueberry.

funders had no role in study design, data collection and analysis, decision to publish or preparation of the manuscript.

**Competing interests:** The authors have declared that no competing interests exist.

# 1 Introduction

Blueberry is a plant of Ericaceae (*Vaccinium* spp.), and its fruit is rich in anthocyanins. Anthocyanin biosynthesis is affected by various factors such as plant growth and development, light, temperature, hormones and enzyme activities [1–3], among which light is one of the critical environmental factors affecting anthocyanin biosynthesis, incredibly light intensity [4, 5].

The path of anthocyanin biosynthesis usually contains an early stage and a late stage. In the early stage, PAL, C4H (cinnamate 4-hydroxylase), 4CL (4-coumarate: CoA ligase), CHS (chalcone synthase), CHI and F3H (flavanone 3-hydroxylase) catalyze the formation of dihydroflavonols. In the late stage, the expression of enzyme genes such as DFR, ANS (anthocyanidin synthase) and UFGT catalyzes the formation of anthocyanins [6, 7]. Studies have found that different light intensities affect enzyme activities by regulating the expression of transcription factors, light signal transduction pathways, phytochrome, cryptochrome and other enzyme genes, which regulated the synthesis of anthocyanins [8, 9]. For example, Zhu et al. [10] clarified that under low light stress, the activities of CHI, CHS and F3H involved in the anthocyanin biosynthesis pathway of purple cabbage decreased, resulting in a decrease in anthocyanin content.

Previous studies have been carried out on the regulation of light intensity on $GA_3$, ETH, JA, IAA, ABA, CTK (cytokinin) and other phytohormones [11, 12], as well as related studies on the effect of hormones on anthocyanin content. For example, reduced the ABA content inhibited anthocyanin synthesis in strawberry fruits [13], and exogenous ABA increased anthocyanin content in rabbiteye blueberry 'Brightwell' [14]. Some studies have also shown that high light can regulate anthocyanin content in wild-type Arabidopsis by stimulating the content of JA [15].

At present, the light intensities regulate blueberry fruits anthocyanin biosynthesis mainly focus on the perspectives of biological enzyme activity and pigment content [16]. However, it is not clear how the light intensity affects the physiological mechanism of anthocyanin content by affecting the endogenous hormones content of blueberry fruits and the associated enzyme activities in the anthocyanin synthesis pathway. Therefore, this study analyzed the changes of anthocyanin content in blueberry fruits under different light intensities at stage S1, S2 and S3, and its correlation with the content of endogenous hormones ($GA_3$, JA, IAA, ABA and ETH) and activities of associated enzymes (PAL, CHI, DFR and UFGT). To explore the correlation between anthocyanin content and endogenous hormones content and enzyme activities under different light intensities, and to explore the regulation mechanism of anthocyanin content in blueberry fruits under different light intensities from the physiological level, to provide a scientific basis for artificial regulation of anthocyanin content in production.

# 2 Materials and methods

## 2.1 Overview of the experimental site

The experimental site is located in the Experimental Nursery of College of Forestry, South Campus of Guizhou University, Huaxi District, Guiyang, with an altitude of 1159 m, 104°34' east longitude, and 26°34' North latitude. It is a subtropical humid and moderate climate. The maximum temperature is 39.5°C, the minimum temperature is -9.5°C, and the average annual temperature is 15.8°C. The yearly effective accumulated temperature above 10°C is 4637.5°C, the annual precipitation is 1229 mm, the annual average relative humidity is 79%, and the total integrated solar radiation is 3567 MJ/$m^2$.

## 2.2 Experimental materials

Four-year-old southern highbush blueberry variety 'O'Neal' with the same maturity and growth was used as the experimental material, and the test seedlings were transplanted into

plastic flower pots (The inner diameter is 26.5 cm, the bottom diameter is 17.5 cm, and the height is 19.7 cm). One seedling per pot was cultured with pine forest humus as the substrate. The nutrient content of the substrate is high, with a pH of about 4.8, which can satisfy the normal growth of blueberries, and weeding and irrigation are carried out regularly.

## 2.3 Experiment design

As shown in Table 1, four light intensities were 100% (CK group, shading rate, 0%), 75% (shading rate, 25%), 50% (shading rate, 50%) and 25% (shading rate, 75%) full light intensity, and which were controlled by a photometer and black sunshade net of different densities with 2, 3, 4, 6 and 8 needles. The trial began after the blueberries had bloomed (1 April 2020).

## 2.4 Sample collection

After one month of treatment, according to the test scheme, blueberry plants with consistent growth and normal fruit yield were selected for sample collection. Then random sampling in the group at three fruit development stages (Fig 1) 28 days (white fruit stage, S1), 35 days (purple fruit stage, S2) and 42 days (blue fruit stage, S3) after full bloom, and pick fruits with uniform size and basically the same size in the middle of the canopy around the tree. 10 g biological replicates were sampled from each biological replicates in each stage, and a total of 3 biological replicates were used for experimental research. The samples were placed in a screw tip bottom centrifuge tube wrapped with tin foil paper, stored in liquid nitrogen, and returned to the laboratory for storage in an ultra-low temperature refrigerator at -80°C.

## 2.5 Method of index determination

**2.5.1 Methods for determination of endogenous hormones content and enzyme activities.** The content of endogenous hormone and the activities of associated enzymes in anthocyanin synthesis pathway were determined by double antibody sandwich enzyme-linked immunosorbent assay (ELISA) [17]. The collected blueberry fruits were tested using an ELISA kit produced by Guizhou Wela Technology Limited Liability Company. Sample treatment: The tissue was rinsed with pre-cooled PBS (0.01M, pH = 7.4), and the weighed 0.1 g fruit and the corresponding volume of PBS (according to the weight to volume ratio of 1:9) were added to the homogenizer for grinding. To further lyse the tissue cells, the homogenate was broken by ultrasound. Finally, the homogenate was centrifuged at 10000 rpm for five minutes, and the supernatant was taken for detection. The content of gibberellin 3 ($GA_3$), jasmonic acid (JA), indoleacetic acid (IAA), abscisic acid (ABA) and ethylene (ETH), and the activities of

**Table 1. Actual light intensity corresponding to relative light intensity.**

| Light intensity | S1/umol·m$^{-2}$s$^{-1}$ | S2/umol·m$^{-2}$s$^{-1}$ | S3/umol·m$^{-2}$s$^{-1}$ |
|---|---|---|---|
| 25% | 372±34.06Ad | 369±29.44Ad | 379±28.29Ad |
| 50% | 750±31.18Ac | 699±24.83Ac | 778±30.02Ac |
| 75% | 1123±40.99Ab | 1094±48.50Ab | 1143±36.37Ab |
| CK | 1498±39.26Aa | 1456±44.46Aa | 1587±37.53Aa |

Note: The above table shows the light intensity at 10 a.m. and is measured with a photometer. S1: white fruit stage, S2: purple fruit stage, S3: blue fruit stage. In the table, different uppercase letters indicate significant differences in the same light intensity during different stages, and different lowercase letters indicate significant differences in different light intensity treatments at the same stage ($P < 0.05$), values represent mean ± standard error.

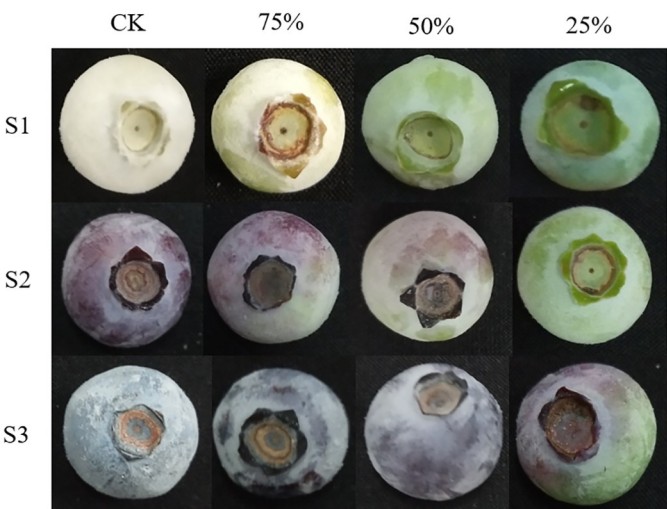

**Fig 1. Blueberry fruits at different development stages under different light intensities.** Note: CK: 100% light intensity, 75%: 75% light intensity, 50%: 50% light intensity, 25%: 25% light intensity; S1: white fruit stage, S2: purple fruit stage, S3: blue fruit stage.

phenylalanine ammonia lyase (PAL), chalcone isomerase (CHI), dihydroflavonol reductase (DFR) and UDP-glucose: flavonoid 3-glucosyltransferase (UFGT) were detected.

**2.5.2 Method for determination of anthocyanin content.** The pH differential method [18, 19] was used to determine the anthocyanin content of whole blueberry fruits: ①0.5 g of blueberry fruit samples of each stage and treatment were weighed and ground into powder in liquid nitrogen, and extracted with 1%HCL-CH$_3$OH (5 mL). ②Take 0.1 mL of the extract in step ①, add KCL buffer (pH = 1) and NaAC buffer (pH = 4.5) 4.9 mL, respectively, and stabilize in a water bath at 40˚C for 20 min. ③Transfer the solution in step ② to a 1mL light path cuvette, and use distilled water as a blank control. Each sample was repeated three times with 10 uL injection each time, colorimetric and absorbance values were read at 520 nm and 700 nm.

Anthocyanin content was calculated according to formulas (1) and (2).

Anthocyanin concentration:

$$C = A*M*N/\varepsilon L \ (mg/mL) \tag{1}$$

Where, A: $[(A_{520}-A_{700})_{pH=1.0}-(A_{520}-A_{700})_{pH=4.5}]$, M: 449.2 (g/mol; relative molecular mass of cyanidin-3-glucoside), N: dilution multiple, $\varepsilon$: 26900 (L/cm/mol; extinction coefficient of cyanidin-3-glucoside), L: optical path length (1cm).

Anthocyanin content:

$$C*V/m \ (mg/g) \tag{2}$$

Where, C: anthocyanin concentration (mg/mL), V: extraction volume (mL), m: blueberry fresh weight (g).

## 2.6 Data analysis

Excel 2019 and Origin 2022 were used for sorting, calculating, mapping data and correlation analysis. One-way ANOVA and Tukey's test were performed using SPSS 19.0. Statistical differences were marked by sequential letter labeling.

## 3 Results

### 3.1 Effect of light intensity on endogenous hormones content in blueberry fruits

As shown in Fig 2, the endogenous hormones content in blueberry fruits was significantly different with different developmental stages and light intensities ($P < 0.05$). With the increase of fruit development and light intensity, the content of $GA_3$ and IAA decreased gradually, in contrast to that the content of JA, ABA and ETH increased gradually.

The effects of different light intensities on the content of $GA_3$ and IAA in blueberry fruits are shown in Fig 2a and 2b. Except for the content of IAA at S3 under 50% light intensity, the content of $GA_3$ and IAA in fruits under 25% light intensity from S1 to S3 were significantly

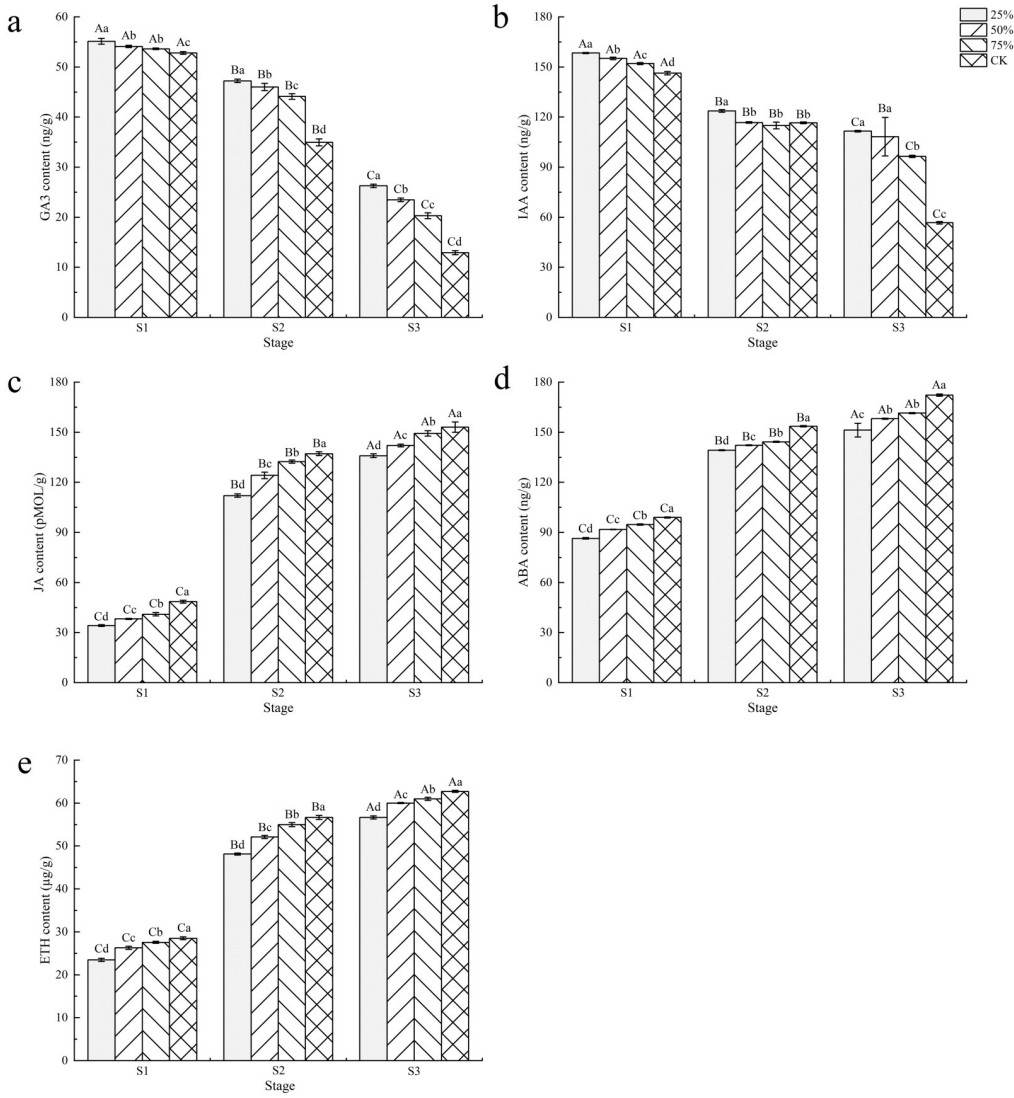

**Fig 2. Effect of light intensity on endogenous hormones content in blueberry fruits.** Note: In the figure, different uppercase letters indicate significant differences in the same light intensity during different stages, and different lowercase letters indicate significant differences in different light intensity treatments during the same stage ($P < 0.05$). Bars show standard deviation.

higher than those under other light intensity treatments at the same stage. Under the same light intensity, the content of $GA_3$ and IAA in S1 were significantly higher than those of S2 and S3 ($P < 0.05$). Under 25% light intensity, the $GA_3$ content of fruits of S1 was 4.41%, 2.80% and 1.87% higher than that of CK, 75% and 50% light intensities at the same stage, respectively. S2 increased by 35.15%, 7.07% and 2.70% respectively, and S3 increased by 103.15%, 29.44% and 11.84% respectively. From S1 to S3, the IAA content of fruits under 25% light intensity was 1.02~1.08 times, 1.06~1.08 times and 1.03~1.97 times of that under other light intensity treatments, respectively. The results indicated that the higher the light intensity, the more inhibited the synthesis of $GA_3$ and IAA in blueberry fruits.

The effects of different light intensities on the content of JA, ABA and ETH in blueberry fruits are shown in Fig 2c–2e. From S1 to S3, the content of JA, ABA and ETH in fruits of CK treatment were significantly higher than those of other shading treatments at the same stage. Under the same light intensity, the content of JA, ABA and ETH in fruits in S3 were significantly higher than those of S1 and S2 ($P < 0.05$). Compared with CK, the JA content of fruits at stage S1 decreased by 15.53%~29.49%, with an average decrease of 22.11%. In S2, the decrease was 3.46%~18.27%, with an average decrease of 10.39%. In S3, the decrease is 2.43%~11.17%, and the average decrease is 6.93%. The content of ABA and ETH at different developmental stages also showed similar changes under different light intensities. The results showed that the increase of light intensity and fruits maturity were beneficial to the synthesis of the JA, ABA and ETH content in blueberry fruits, but the difference between the shading treatment and CK decreased with the increase of fruits maturity.

## 3.2 Effect of light intensity on associated enzyme activities in the anthocyanin biosynthesis pathway of blueberry fruits

The effects of different light intensities on the activities of associated enzymes in the anthocyanin synthesis pathway of blueberry fruits are shown in Fig 3. The activities of PAL, CHI, DFR and UFGT in blueberry fruits gradually increased with growth and development and light intensity, and the fruit development stage and light intensity had significant effects on them ($P < 0.05$).

Among the four enzyme activities in different stages, except that there was no significant difference in the activity of PAL between 75% light intensity treatment and CK treatment at stage S3, the enzyme activity of CK treatment in other stages was significantly higher than that of shading treatment at the same stage ($P < 0.05$). At the same light intensity, the activities of the four enzymes in the fruits at S3 were significantly higher than those of S1 and S2 ($P < 0.05$). Compared to CK, the PAL activity of fruits in each shading treatment decreased by 6.83% ~ 17.12% at stage S1, with an average decrease of 12.84%. It decreased by 22.40% ~ 32.08% and 27.21% on average at stage S2. It decreased by 2.41% ~ 32.65% at stage S3, and decreased by 15.09% on average. The activities of CHI, DFR and UFGT in different stages of fruit development under different light intensities also had similar changes. It can be seen that the activities of four enzymes in the anthocyanin synthesis pathway of blueberry fruits were positively correlated with light intensity and growth.

## 3.3 Effect of light intensity on anthocyanin content in blueberry fruits

The anthocyanin content of blueberry fruits was significantly affected by light intensity and fruit maturity ($P < 0.05$, Fig 4). The anthocyanin content of blueberry fruits in the same development stage gradually increased with the increase of light intensity, and the anthocyanin

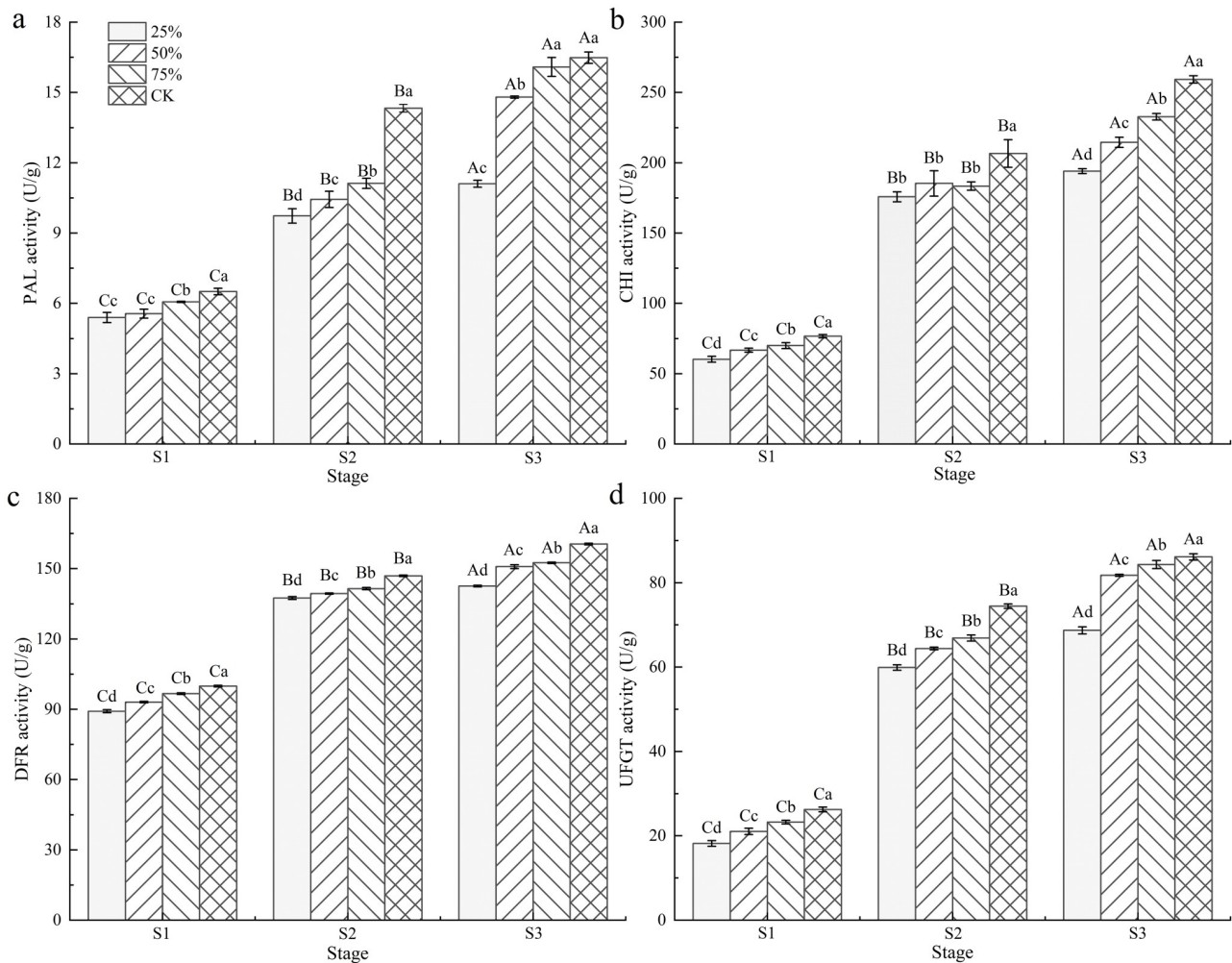

**Fig 3. Effect of light intensity on associated enzyme activities in the anthocyanin biosynthesis pathway of blueberry fruits.** Note: In the figure, different uppercase letters indicate significant differences in the same light intensity during different stages, and different lowercase letters indicate significant differences in different light intensity treatments during the same stage ($P < 0.05$). Bars show standard deviation.

content of blueberry fruits under the same light intensity also gradually increased with the development of fruits.

From S1 to S3, the anthocyanin content of fruits under CK treatment was significantly higher than that under shading treatment ($P < 0.05$). There was no significant difference in anthocyanin content among the three shading treatments at stage S1, which may be due to the less synthesis of anthocyanin and the relatively small effect of light intensity on anthocyanin content at stage S1. The anthocyanin content of fruits treated with CK at stage S2 was 0.453 mg/g, which was 4.91 times, 6.23 times and 24.13 times of that under 75%, 50% and 25% light intensity at the same stage, respectively. The anthocyanin content of fruits under CK treatment in S3 was as high as 1.254 mg/g, which was 3.24 times, 6.68 times and 14.91 times of 75%, 50% and 25% light intensity treatments at the same stage, and 30.32 times and 2.77 times of CK treatment at S1 and S2, respectively. The results showed that shading was not conducive to anthocyanin synthesis in blueberry fruits, which was consistent with the phenotype that the higher the light intensity, the darker the appearance color of blueberry fruits.

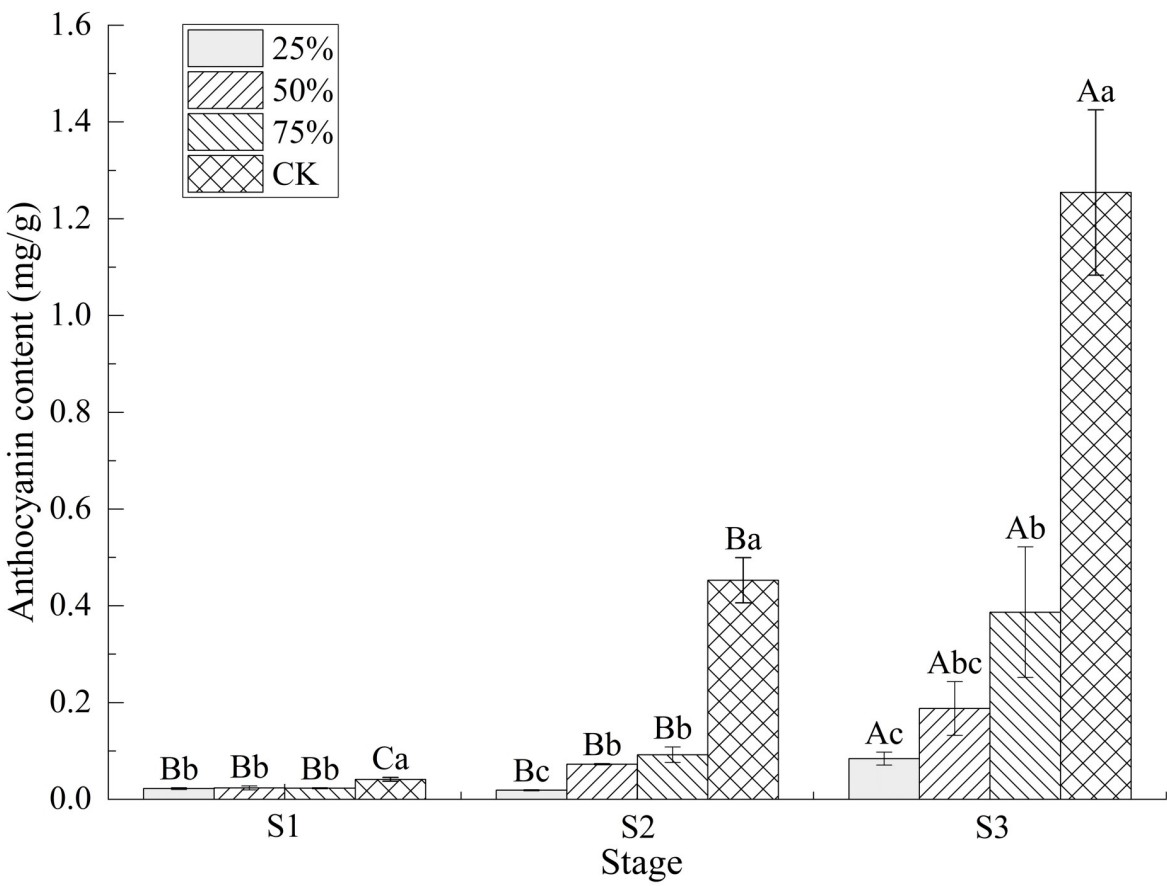

**Fig 4. Effect of light intensity on anthocyanin content in blueberry fruits.** Note: In the figure, different uppercase letters indicate significant differences in the same light intensity during different stages, and different lowercase letters indicate significant differences in different light intensity treatments during the same stage ($P < 0.05$). Bars show standard deviation.

### 3.4 Correlation analysis of anthocyanin content with light intensity, endogenous hormones and enzyme activities in blueberry fruits

The correlation analysis of anthocyanin content with light intensity, endogenous hormones (GA₃, JA, IAA, ABA, ETH) and activities of associated enzymes in the anthocyanin synthesis pathway (PAL, CHI, DFR, UFGT) in blueberry fruits at different developmental stages under different light intensity conditions is shown in Fig 5. It can be seen that the anthocyanin content of blueberry fruits at three development stages was significantly ($P < 0.05$) or extremely significantly ($P < 0.01$) positive correlated with light intensity, and the correlation gradually increased with the development of fruits. The anthocyanin content of blueberry fruits at three developmental stages was negatively correlated with the content of GA₃ and IAA, and positively correlated with the other three hormones content and four enzyme activities, which was consistent with the trend of anthocyanin content, endogenous hormones content and associated enzyme activities in the anthocyanin synthesis pathway with light intensity in this study.

The anthocyanin content of blueberry fruits under different light intensities was significantly negative correlated with the content of GA₃ at stage S1 ($P < 0.05$), was highly significantly negative correlated with the content of IAA ($P < 0.01$), was extremely significantly positive correlated with the content of JA, as well as the activities of PAL, CHI and UFGT

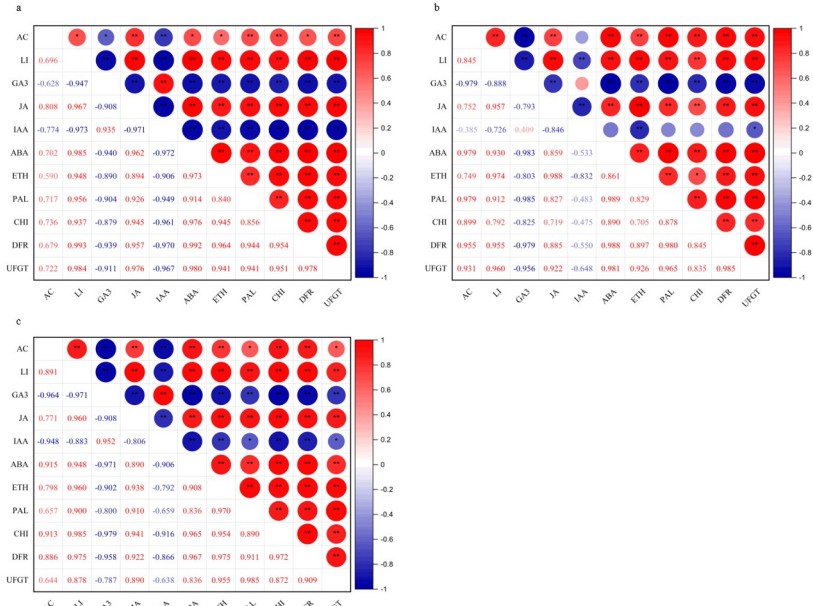

**Fig 5. Correlation analysis of anthocyanin content with light intensity, endogenous hormones and enzyme activities in blueberry fruits at different developmental stages under light intensity treatment.** Note: AC: anthocyanin content, LI: light intensity; a: white fruit stage (S1); b: purple fruit stage (S2), c: blue fruit stage(S3); * indicates significant correlation at 0.05 level, ** indicates highly significant correlation at 0.01 level.

($P < 0.01$), and was significantly positive correlated with the content of ABA, ETH and the activity of DFR ($P < 0.05$). At stage S2, it was extremely significantly negative correlated with the content of $GA_3$ ($P < 0.01$), and was extremely significantly positive correlated with the content of JA, ABA and ETH and the activities of PAL, CHI, DFR and UFGT ($P < 0.01$), and the correlation coefficients with the content of $GA_3$, ABA and the activity of PAL were 0.979. At stage S3, it was highly significantly negative correlated with the content of $GA_3$ and IAA ($P < 0.01$), significantly positive correlated with the activities of PAL and UFGT ($P < 0.05$), and had a highly significantly positive correlation with the activities of the other three hormones and two enzymes ($P < 0.01$). The results showed that the regulation of light intensity on anthocyanin synthesis in blueberry fruit was closely related to five endogenous hormones and four associated enzyme activities in the anthocyanin synthesis pathway.

## 4 Discussion

Plant hormones play an important role in plant growth and development and plant response to environmental stress. Studies showed that shading increased the content of $GA_3$ and IAA, and decreased the content of ABA in peony seed oil [20]. Shading can promote the accumulation of GA in sunflower [21]. Adequate light can promote JA biosynthesis in Arabidopsis seedlings [22]. In this study, the content of $GA_3$ and IAA in blueberry fruits decreased gradually with the increase of light intensity, while the content of JA, ABA and ETH increased gradually with the increase of light intensity, which was consistent with literature reports. However, Gao et al. [23] found that shading significantly reduced the IAA content in summer maize grains and increased the ABA content, which may be due to the physiological differences of maize as a C4 plant, or may be due to the different effects of light intensity on different plants. It was found that exogenous ABA promoted anthocyanin accumulation in tomato hypocotyls, while

GA inhibited anthocyanin accumulation [24]. IAA can inhibit anthocyanin biosynthesis in *Arabidopsis thaliana* red *pap1-D* calli [25]. JA can promote the synthesis of anthocyanins in grapes [26]. ABA can stimulate anthocyanin accumulation during fruit ripening of 'Jersey' highbush blueberry [27]. Exogenous ETH treatment significantly increased anthocyanin accumulation in plum peel [1] and grape berries [28]. This is consistent with the change trend of endogenous hormones content and anthocyanin content with light intensity in this study.

During the ripening of sweet cherry, the content of IAA was significantly negative correlated with the anthocyanin content ($P < 0.05$), the content of $GA_4$ and ABA were highly significantly correlated with the anthocyanin content ($P < 0.01$), the former was negatively correlated, the latter was positively correlated [29], which was consistent with the results of this study. The content of IAA was positively correlated with the anthocyanin content during the development of bicolor kale leaves [30], which is contrary to the results of this study, and may be caused by differences in plant varieties and organs. In conclusion, light intensity affects anthocyanin synthesis in blueberry by regulating the content of endogenous hormones, but there are significant differences in the types of hormones in different species.

Many studies have shown that the expression of associated enzyme genes in the anthocyanin synthesis pathway is induced by strong light, thereby regulating its enzyme activity and promoting anthocyanin accumulation. However, under low light or dark conditions, the expression of associated genes was down-regulated or inhibited, and the enzyme activity was reduced, resulting in a decrease in anthocyanin accumulation in flowers and fruits [31–34]. Dong et al. [35] found that blocking natural light before apple bud breaking reduced the expression of 6 enzyme genes (*PAL*, *CHS*, *CHI*, *F3H*, *DFR* and *ANS*) and inhibited anthocyanin biosynthesis. The activity of DFR in 'Fuji' apple peel increased with the increase of light intensity, and the synthesis of anthocyanin was regulated by DFR activity [36]. The lack of anthocyanin in Matthiola line white flowers [37] and white grapes [38] is due to the lack of activities of DFR and UFGT, respectively.

Under different light intensities, the activities of PAL, CHI and DFR in the leaves of *Betula* Royal Frost were highly significantly positive correlated with the anthocyanin content ($P < 0.01$) [39], and the correlation between *DFR*, *UFGT* genes and anthocyanin content in strawberry fruit reached a significant level [40] ($P < 0.05$), which were consistent with the results of this study. With the development of blueberry fruits and the enhancement of light intensity, the activities of PAL, CHI, DFR and UFGT in blueberry fruits in this study showed an increasing trend, which was consistent with the change trend of anthocyanin content in blueberry fruits, and there was a significant or extremely significantly positive correlation between the activities of the four enzymes and anthocyanin content. It can be seen that light intensity affects anthocyanin synthesis by regulating associated enzyme activities in the anthocyanin synthesis pathway.

In addition, during the stage S2, the anthocyanin content of blueberry fruits under different light intensities was extremely significantly negative correlated with the $GA_3$ content ($P < 0.01$), was extremely significantly positive correlated with the JA, ABA, ETH content and PAL, CHI, DFR, UFGT activity ($P < 0.01$), and the correlation coefficient was high, but it did not reach a significant level with the IAA content, and the correlation coefficient was 0.385. This may be because the purple fruit stage is the period when anthocyanins start to be synthesized and accumulated in large quantities in blueberry fruits, and the endogenous hormones content and associated enzyme activities that positively affect anthocyanin content biosynthesis are higher, which provide material basis for the synthesis of anthocyanins.

The synthesis of anthocyanins in most plants are related to light intensity, which have been reported in apple [41], strawberry [42], litchi [5], grape [43], *Betula* Royal Frost [39] and other plants. Shao et al. [42] found that the anthocyanin accumulation of strawberry under 25% and

75% light intensity treatment was lower than that under 100% light intensity treatment. The red grape variety 'Jingxiu' cannot synthesize and accumulate anthocyanins in the absence of light [31]. Yan et al. [44] found that the anthocyanin content in fruits of four rabbiteye blueberry varieties, namely 'Powderblue', 'Brightwell', 'Baldwin' and 'Tifblue', increased with the increase of light intensity when different shading treatments were applied. Guo et al. [45] also conducted a similar experimental treatment on rabbiteye blueberry 'Powderblue', and verified again that the anthocyanin content of its fruit increased with the increase of light intensity. Some studies have also shown that anthocyanin synthesis during apple flower development is controlled both by growth and light intensity [35]. The results of this study confirmed that synthesis the anthocyanin in blueberry fruits were also significantly affected by light intensity.

## 5 Conclusion

The synthesis of anthocyanins is related not only to its own genetic factors but also to many external environmental factors. Pearson's correlation analysis showed that five hormones and four enzyme activities were significantly or extremely significantly correlated with the anthocyanin content in blueberry fruits. Among them, the JA content and the CHI activity had significant effects on the synthesis of anthocyanins at different stages of development. The results showed that the light intensity affected the synthesis of anthocyanins by regulating the content of endogenous hormones and the activity of associated enzymes in the anthocyanin synthesis pathway. The more sufficient light, the more conducive the synthesis of anthocyanins in blueberry. These findings are foundational for further research on the mechanism of light intensity-induced anthocyanin synthesis in blueberry.

## Supporting information

**S1 Dataset.**
(XLSX)

## Author Contributions

**Data curation:** Xiaoli An, Zejun Song.

**Formal analysis:** Xiaoli An, Xiaolan Guo, Xinyu Zhang, Yunzheng Zhu.

**Funding acquisition:** Delu Wang.

**Investigation:** Tianyu Tan.

**Methodology:** Delu Wang.

**Resources:** Delu Wang.

**Visualization:** Xiaoli An.

**Writing – original draft:** Xiaoli An.

**Writing – review & editing:** Delu Wang.

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
