## [Decision Letter · Decision Letter 0]

5 Feb 2023

PONE-D-22-30734Physiological response of anthocyanin synthesis and accumulation to light intensity in blueberryPLOS ONE

Dear Dr. Wang,

Thank you for submitting your manuscript to PLOS ONE. After careful consideration, we feel that it has merit but does not fully meet PLOS ONE’s publication criteria as it currently stands. Therefore, we invite you to submit a revised version of the manuscript that addresses the points raised during the review process.

We look forward to receiving your revised manuscript.

Kind regards,

Zhenhai Han, PhD

Academic Editor

PLOS ONE

Journal Requirements:

“Yes.

Delu Wang, grant number 31760205,National Natural Science Foundation of China.”

Additional Editor Comments:

Reviewer #1:

In my opinion, this article is scientifically valuable and can be accepted with some revisions.

1- Quantify the abstract part of the article by adding the percentage of reduction and increase of studied traits.

2- Add complete explanations to the materials and methods section about plant growth conditions (environmental temperature during plant growth, the characteristics of the growth medium such as the content of nutrients, pH, cation exchange capacity, soil texture, plant nutrition conditions, irrigation, etc).

3. In figures: Add information about the bars on the columns to the captions of the figures (Bar type? SD or SE?).

Reviewer #2:

1. Title

• The title is not adequate in terms of description and declaration but “I recommend to add varying or different light intensity” and also rephrase it because after synthesis there is accumulation I think this repetition.

2. Abstract

• The abstract is clear and straight to the point and gives a clear summary of the study.

3. Introduction

• The introduction of this manuscript is clear as it motivates and the objective of the study is concise and to the point.

4. Materials and Methods

• This section is well written with clear steps that are precise

• The procedure and material used are relevant to the study.

• The equipment used are also well described

• However, some section is not clear in terms of language flow. For instance, Lines 102-104 "After one month of treatment, according to the test scheme, 5 to 8 blueberry plants with consistent growth and normal fruit yield, were as the source of a biological duplicate sample. "

5. Results

• The results are clear with adequate pictures, figures, tables, graphs and supplementary materials

• The sentence flow is not so well written and it is not easy to understand. For instance, Lines 162-163 ". In contrast, the content of JA, ABA and ETH increased with the growth and development and the light intensity increased."

6. Discussion

• The discussion is also very adequate and relevant for this study, it well cited with current research relevant to this study.

• There is no need of subtitles in the discussion part i.e 4.1, 4.2.

7. Conclusion

• The conclusion is not very conclusive because it does not show what is the next step or direction of research.

8. References

• The references are written well and there are consistent through the text

9. Tables and Figures

• The tables and figures in this draft manuscript are emphasized accordingly. However, Figures 2, 3 and 4 are not very visible.

• The figure and tables are also clearly labeled

10. Supplementary materials

• There are no supplementary materials to support the findings results of this research

11. General comments

• The manuscript is adequate as it has tried to address the relationship between different light intensity and anthocyanin biosynthesis and accumulation in respect to hormones and the activity of various enzyme involved.

• This draft manuscript meets minimum objective of research in terms of materials and methods, results, discussion and conclusion, the figures and tables are also well organized. A little improvement is necessary as per recommendation.

• I would recommend further editing of the manuscript so as to improve the manuscript and then it can be accepted.

Reviewers' comments:

Reviewer's Responses to Questions

**Comments to the Author**

1. Is the manuscript technically sound, and do the data support the conclusions?

Reviewer #1: Yes

2. Has the statistical analysis been performed appropriately and rigorously? 

Reviewer #1: Yes

3. Have the authors made all data underlying the findings in their manuscript fully available?

Reviewer #1: Yes

4. Is the manuscript presented in an intelligible fashion and written in standard English?

Reviewer #1: Yes

5. Review Comments to the Author

Reviewer #1: In my opinion, this article is scientifically valuable and can be accepted with some revisions.

1- Quantify the abstract part of the article by adding the percentage of reduction and increase of studied traits.

2- Add complete explanations to the materials and methods section about plant growth conditions (environmental temperature during plant growth, the characteristics of the growth medium such as the content of nutrients, pH, cation exchange capacity, soil texture, plant nutrition conditions, irrigation, etc).

3. In figures: Add information about the bars on the columns to the captions of the figures (Bar type? SD or SE?).

6. PLOS authors have the option to publish the peer review history of their article (what does this mean?). If published, this will include your full peer review and any attached files.

Reviewer #1: **Yes: **Salar Farhangi-Abriz

---

## [Author Response · Author response to Decision Letter 0]

10 Feb 2023

Dear Editor and reviewers:

Thank you very much for your useful comments and professional advice on our manuscript. We wish to give a sincere gratitude to referees for reviewing our article carefully. These opinions help to improve academic rigor of our article, and we apologize for any inconveniences caused by these errors. Based on your advice and requests, we have modified the manuscript accordingly, and the response to the referees’ comments are listed point by point below:

Reviewer #1:

Comment 1: Quantify the abstract part of the article by adding the percentage of reduction and increase of studied traits.

Reply 1: Thanks for your comment and suggestion. Your suggestion plays a vital role in improving the quality of our paper. We have added partial percentage data of studied traits to the abstract to make it more convincing.

Comment 2: Add complete explanations to the materials and methods section about plant growth conditions (environmental temperature during plant growth, the characteristics of the growth medium such as the content of nutrients, pH, cation exchange capacity, soil texture, plant nutrition conditions, irrigation, etc).

Reply 2: We feel sorry that we did not provide enough complete information about plant growth conditions in the Materials and Methods section. First of all, the environmental temperature during plant growth has been explained in 2.1. Secondly, as for the characteristics of the growing medium, we are very sorry for the lack of detailed data such as soil nutrient content, cation exchange capacity and soil texture. This is a mistake of our research, and we will be more cautious in future research. But our substrate is purchased from professional blueberry growing base with high nutrient content and pH of about 4.8, which can satisfy the normal growth of blueberries. This has been supplemented accordingly in the manuscript. In addition, the management measures of blueberry plants were consistent in our study, except for the inconsistent light intensity gradient treatment.

Comment 3: In figures: Add information about the bars on the columns to the captions of the figures (Bar type? SD or SE?).

Reply 3: We are grateful for the suggestion. The bars in Figures 2 ~ 4 are all standard deviations, which we have added in the notes of the corresponding figures.

Reviewer #2:

 Comment 1: Title

• The title is not adequate in terms of description and declaration but “I recommend to add varying or different light intensity” and also rephrase it because after synthesis there is accumulation I think this repetition.

Reply 1: Thank you for your careful review. Your suggestion is of great help to improve the quality of our paper. According to your suggestion for our title, we have corrected the “Physiological response of anthocyanin synthesis and accumulation to light intensity in blueberry” into “Physiological response of anthocyanin synthesis to different light intensities in blueberry”.

Comment 2: Abstract

• The abstract is clear and straight to the point and gives a clear summary of the study.

Reply 2: Thank you for your comment, and I will carry out my future research more seriously with your encouragement.

Comment 3: Introduction

• The introduction of this manuscript is clear as it motivates and the objective of the study is concise and to the point.

Reply 3: Thank you for your comments and affirmation. I will work harder in the future research.

 Comment 4: Materials and Methods

• This section is well written with clear steps that are precise.

• The procedure and material used are relevant to the study.

• The equipment used are also well described.

• However, some section is not clear in terms of language flow. For instance, Lines 102-104 "After one month of treatment, according to the test scheme, 5 to 8 blueberry plants with consistent growth and normal fruit yield, were as the source of a biological duplicate sample. "

Reply 4: Thank you for your comments, suggestions and appreciation. Some section of our article is not clear in terms of language flow, we have made corresponding modifications in the manuscript, and we hope to get your approval.

Comment 5: Results

• The results are clear with adequate pictures, figures, tables, graphs and supplementary materials.

• The sentence flow is not so well written and it is not easy to understand. For instance, Lines 162-163 ". In contrast, the content of JA, ABA and ETH increased with the growth and development and the light intensity increased."

Reply 5: First of all, thank you for your comments and affirmation, which is undoubtedly a strong support for our research work. Secondly, thank you for your advice. We are sorry that our sentence flow is not well written, making it difficult to understand and increasing your workload. Meanwhile, we have made corresponding revisions in the manuscript.

Comment 6: Discussion

• The discussion is also very adequate and relevant for this study, it well cited with current research relevant to this study.

• There is no need of subtitles in the discussion part i.e 4.1, 4.2.

Reply 6: Thank you for your comments, appreciation and suggestions. As for the subtitle of the discussion section, we fully agree with your suggestion and have deleted it in the manuscript.

Comment 7: Conclusion

• The conclusion is not very conclusive because it does not show what is the next step or direction of research.

Reply 7: Thank you for your comment and suggestion, which will greatly help to improve the quality of our paper. Our study laid a foundation for further study on the mechanism of light intensity-induced anthocyanin synthesis in blueberry, and we have supplemented it in the manuscript.

Comment 8: References

• The references are written well and there are consistent through the text.

Reply 8: Thank you for your comment and affirmation. I will continue to carry out the next step of research with your affirmation.

Comment 9: Tables and Figures

• The tables and figures in this draft manuscript are emphasized accordingly. However, Figures 2, 3 and 4 are not very visible.

• The figure and tables are also clearly labeled.

Reply 9: Thanks for your comment and suggestion. We have emphasized the figures in Figures 2~4 accordingly in the manuscript. For example, the bars in Figures 2 ~ 4 are all standard deviations, which we have added in the notes of the corresponding figures.

Comment 10: Supplementary materials

• There are no supplementary materials to support the findings results of this research

Reply 10: Thank you for your comments. No supplementary material is available in this article.

Comment 11: General comments

• The manuscript is adequate as it has tried to address the relationship between different light intensity and anthocyanin biosynthesis and accumulation in respect to hormones and the activity of various enzyme involved.

• This draft manuscript meets minimum objective of research in terms of materials and methods, results, discussion and conclusion, the figures and tables are also well organized. A little improvement is necessary as per recommendation.

• I would recommend further editing of the manuscript so as to improve the manuscript and then it can be accepted.

Reply 11: Thank you very much for your comments and suggestions. According to your suggestions, we have revised and supplemented the manuscript accordingly.

---

## [Decision Letter · Decision Letter 1]

6 Mar 2023

Physiological response of anthocyanin synthesis to different light intensities in blueberry

PONE-D-22-30734R1

Dear Dr. Wang,

We’re pleased to inform you that your manuscript has been judged scientifically suitable for publication and will be formally accepted for publication once it meets all outstanding technical requirements.

Kind regards,

Zhenhai Han, PhD

Academic Editor

PLOS ONE

Additional Editor Comments (optional):

Reviewers' comments:

Reviewer's Responses to Questions

**Comments to the Author**

1. If the authors have adequately addressed your comments raised in a previous round of review and you feel that this manuscript is now acceptable for publication, you may indicate that here to bypass the “Comments to the Author” section, enter your conflict of interest statement in the “Confidential to Editor” section, and submit your "Accept" recommendation.

Reviewer #1: All comments have been addressed

2. Is the manuscript technically sound, and do the data support the conclusions?

Reviewer #1: Yes

3. Has the statistical analysis been performed appropriately and rigorously? 

Reviewer #1: Yes

4. Have the authors made all data underlying the findings in their manuscript fully available?

Reviewer #1: Yes

5. Is the manuscript presented in an intelligible fashion and written in standard English?

Reviewer #1: Yes

6. Review Comments to the Author

Reviewer #1: (No Response)

7. PLOS authors have the option to publish the peer review history of their article (what does this mean?). If published, this will include your full peer review and any attached files.

Reviewer #1: **Yes: **Salar Farhangi-Abriz

---

## [Editor Report · Acceptance letter]

14 Jun 2023

PONE-D-22-30734R1 

Physiological response of anthocyanin synthesis to different light intensities in blueberry 

Dear Dr. Wang:

I'm pleased to inform you that your manuscript has been deemed suitable for publication in PLOS ONE. Congratulations! Your manuscript is now with our production department. 

Kind regards, 

on behalf of

Dr. Zhenhai Han 

Academic Editor

PLOS ONE